# AbnormalLog: A Deep Anomaly Detection Method for Log Sequence Data

## Abstract

Anomaly detection for computer log sequence data plays a very important role in various industries. Log data is complex time series with plenty of text information, which is difficult to process due to both its non-structural characteristics and temporal correlation. Existing log anomaly detection schemes do not utilize all available data information such as the semantic and parameter information, nor do they consider weighting of data based on time. The AbnormalLog algorithm proposed in this paper implements semantic parsing technique to expand current detection schemes by analyzing template and parameter information of the log data. AbnormalLog is comprised of four functional modules: Log Parsing, Semantic Embedding, Parameter Anomaly Detection and Template Anomaly Detection. We compare the proposed method to three most commonly used log anomaly detection methods in industry. The results demonstrate that AbnormalLog is superior to the other algorithms with respect to common model evaluation criteria.

## 1 Introduction

The anomaly detection for sequence data has very important and extensive applications in various industry areas. The traditional anomaly detection problem, such as in Gao et al. (2019; 2020), focuses on the time series data that only contains numerical information and aims to study the change in the data generating schemes. However, with the development of large-scale computer servers, the sequential data has evolved from the traditional numerical data to the complicated unstructured data, which contains a large amount of text information and numerical information. One typical example of such data is the log data, which is essentially a time series text data generated from the operating system automatically. The log data is composed of the original content and time stamp of the computer log. It records the detailed event information of the system operation, which can help the system administrator to quickly target problems and find errors efficiently. Therefore, log data is treated as one of the most critical information resources for the anomaly detection tasks of the system. The artificial intelligence for IT operations (AIOps) utilizes the log data which contains key information about the operation and maintenance of the IT system to reduce the need for human intervention which reduces costs. The difficulty of log data processing lies on the non-structural characteristics of log data itself and its temporal correlation.

The traditional log detection methods require a lot of expert experience, and do not take advantage of the temporal nature of the log data. Xu et al. (2009a) proposed an analysis method based on PCA. Schölkopf et al. (2001) proposed the One-Class SVM method. Liu et al. (2008) proposed the IsolationForest algorithm. These methods are essentially looking for outliers in the cluster, which are usually treated as abnormalities. Although these methods perform well in general anomaly detection studies, they have very obvious deficiencies in log exception detection. Firstly, it simply characterizes the log data as a vector, and then detects outliers of the vector. Secondly, some important temporal information is missing, where the time stamp of log is not considered as a feature in the analysis. Vaarandi and Pihelgas (2015) proposed the LogCluster algorithm to detect the anomaly of log sequence by comparing the log to an existing cluster by utilizing the characteristics of log information. However, their method failed in diagnoses the template and temporal characteristics of the logs thus cannot effectively distinguish two significant different logs under the same template. For example, "the running time is 1s" and "the running time is 5000s", these two logs with the

same template but with contexts are quite different. The log template extraction methods, such as Drain (He et al., 2017), Spell (Du and Li, 2016), and MoLFI (Messaoudi et al., 2018), were then developed. Among these work, Drain achieved the best performance and the highest accuracy.

In recent years, log anomaly detection methods under the deep learning framework become more and more popular. As a very representative log detection method in recent years, DeepLog (Du et al., 2017) detect the exceptions in template with respect to both template Key and template Value after extracting the template information of the log sequence. However, DeepLog still has some drawbacks. Firstly, for the Key of the predicted log data template, DeepLog clusters logs only based on their One-Hot Encoding results, and does not fully consider the similarity of the semantic information in different templates. For example, "the running time is" and "the runtime of the procedure is" should express the same meaning on some levels. Therefore in our proposed method, if we can cluster these two logs together by considering the semantic information of the logs, it will produce more accurate anomaly analysis results. Loganomaly (Meng et al., 2019) is another popular method, which addressed the drawback of Deeplog, and extract the semantic information of the template by the weighted average of the semantic information using the positive and negative synonyms. In addition, it also considers the anomaly detection in both sequence and quantity. Although this method takes the template information and semantics information into account at the same time, it still has some drawbacks. Firstly, after extracting the template, it does not use the parameter information of the log template. These parameter information usually contains some critical information of log exceptions. Secondly, the first procedure of their semantic embedding algorithm is to establish a special thesaurus for the positive and negative synonyms, and then assign specific weights to these positive and negative synonyms appeared in the log template according to the thesaurus. For example, "the running time is increasing $\cdots$ " and "the running time is decreasing $\cdots$ ", where "Increasing" and "Decreasing" are a pair of antonyms with key information. In a log sequence, antonyms always appear in the position where the log parameters are. Then the Word2vec technique (Mikolov et al., 2013) was developed to embed the log template to make up for the lack of parameter information. However, this technology has been gradually defeated by Bert (Devlin et al., 2019), which has a very high performance in the nature language processing field in recent years. Extensive professional knowledge from the relevant fields is required in determining the size of the moving window and the embedding of positive and negative synonyms, which reduces the automation possibility of the whole method. RobustLog (Zhang et al., 2019) is another representative technology. Similar to Loganomaly, RobustLog converts each log template into a semantic vector with fixed dimensionality. Through the semantic analysis, this method can identify and process new and similar log events that arise from the constantly generated log statements and parsing errors. However, Robustlog also doesn't utilize the log parameter information sufficiently.

To overcome the drawbacks of the existing deep learning methods, we propose a new log anomaly detection method under the deep learning framework named as AbnormalLog. AbnormalLog makes comprehensive use of the log template information, the parameter information and the semantic information to deeply analyze the log sequence and detect all possible log exceptions through well designed functional modules. We compare the performance of AbnormalLog to three commonly used deep learning methods, which are the unsupervised learning methods DeepLog and LogAnomaly, and the supervised learning method RobustLog on two public data sets, BGL and HDFS. The emperical analysis shows the excellent performance of our proposed method.

## 2 METHOD

### 2.1 MODEL AND NOTATION

The log data consists of the original contents of the log and the timestamp, which is essentially a time series composed of text information. The AbnormalLog model treats the streaming log data as a text time series data, and analyze it in combination of the natural language processing technology and time series anomaly detection technology. Suppose $S = \{X_{t-k} | k \in \mathbb{Z}^+, 0 \leq k \leq s - 1\}$ be a log data stream generated from time $t - s + 1$ to $t$ by the operating system. Within the entire log sequence $S$, the abnormal state $Z_t$ of the log $X_t$, which is the log generated at the $t^{th}$ moment, is

$$Z_t = \mathcal{G}\left\{ Z_t^T \cup Z_t^P \right\},$$

$$Z_t^T = G_{\theta_T}(S^T), \text{ and } Z_t^P = G_{\theta_P}(S^P),$$

where, $\mathcal{I}\{\cdot\}$ is an indicator function. $S^T = \{X_{t-s+1}^T, \ldots, X_t^T\}$ and $S^P = \{X_{t-s+1}^P, \ldots, X_t^P\}$ are the template information and parameter information of $S$, $s \in \mathbb{Z}^+$ is the size of the data processing window. $G_{\theta_T}(\cdot)$ is the Template Anomaly Detection module with parameter set $\theta_T$, and $G_{\theta_P}(\cdot)$ is the Parameter Anomaly Detection module with parameter set $\theta_P$. $Z_t^T$, $Z_t^P$ and $Z_t$ are the corresponded template anomaly state, parameter anomaly state, and overall anomaly state of the log data $X_t$ at time $t$. "0" represents normal and "1" represents abnormal.

---

**Algorithm 1:** AbnormalLog

**Input:** The Log streaming data $(X_{t-s+1}, \ldots, X_{t-k}, \ldots, X_t)$ at time $t$, where $0 \le k \le s-1$, and $k, s \in \mathbb{Z}^+$

**Step 0: Create a log template sematic vector set**: map a log templates set $\Lambda^T = \{X_1^T, \ldots, X_n^T\}$ into a log template semantic vector set $\Omega^T = \{(X_i^T, \phi_i)|i = 1, \ldots, n\}$ based on the sentence-bert technique as shown in Section 2.3.1, where $n$ is the total number of templates found in log data.

**Step 1: Log parsing** parse each log data $X_{t-k}$ using Drain to get the their and templates and parameters information $\{X_{t-k}^T, X_{t-k}^P\} = \text{Drain}(X_{t-k})$.

**Step 2: Template anomaly detection**

    **Step 2.1: Semantic embedding**: according to $\Omega^T$, map each log template $X_{t-k}^T$ obtained in Step 1 into a semantic vector $\phi_{t-k} = \sum_{i=1}^n \phi_i \times I(X_{t-k}^T = X_i)$.

        If $(X_{t-k}^T, \phi_{t-k}) \notin \Omega^T$, do

            - include $X_{t-k}^T$ into the log template set $\Lambda^T$;

            - repeat Step 0 to update $\Omega^T$;

            - return back to Step 2.

    **Step 2.2: Template anomaly status evaluation for the target log**:

$$Z_t^T = G_{\theta_T}(\{\phi_{t-k}|k = 0, \ldots, s-1\}).$$

**Step 3: Parameter anomaly detection**: obtain the anomaly status of the parameters for the target log

$$Z_t^P = G_{\theta_P}(\{X_{t-k}^P|k = 0, \ldots, s-1\}).$$

**Step 4: Obtain the overall anomaly status of the target log**:

$$Z_t = \mathcal{I}\left\{ Z_t^T \cup Z_t^P \right\}.$$

**Output:** The anomaly status $Z_t$ of the target log $X_t$.

---

Algorithm 1 is the computational process of our proposed AbnormalLog method. AbnormalLog consists of four functional modules, which are Log Parsing, Semantic Embedding, Parameter Anomaly Detection and Template Anomaly Detection. Since the template part and parameter part of the log data provide different level of semantic information, it is necessary to detect the exception status of the template and parameter separately. The next challenge is how to design the corresponding anomaly detection scheme for the template and parameters. Usually, the anomaly log detected by the algorithm needs to be checked manually. In practice, people always have different definitions of exceptions based on their own perceptions. Therefore, in the process of model training, we need to establish different annotation schemes to adapt to different scenes. The computational workflow of the AbnormalLog algorithm is summarized as follows. First, the log sequence at time $t$ is parsed to obtain all template and parameter information. Then the abnormality detection of the template and parameter are carried out simultaneously. Finally, whether the log sequence generated at time $t$ is abnormal or not, will be determined by the results of the template anomaly detection model and the parameter anomaly detection model through the indicator function $\mathcal{I}\{\cdot\}$.

115 ## 2.2 LOG PARSING

116 For log streaming data, because the log parameters and templates are of slightly different importance,
117 it is necessary to design different anomaly detection schemes for these two parts. In the log parsing
118 phase, we use Drain to separate the template and the parameter information. For log streaming data,
119 because the log parameters and templates are of slightly different in importance, it is very necessary
120 to design different anomaly detection schemes for these two parts in algorithm construction. In the
121 log parsing phase, we use Drain (He et al., 2017) to separate the template and parameter information.
122 That is

$$\{X_t^T, X_t^P\} = \text{Drain}(X_t),$$

123 where $\text{Drain}(\cdot)$ is the log parse tree of Drain with fixed depth. Its performance has achieved SOTA.
124 The template and the parameter information can be obtained well through log parse tree constructed
125 from Drain.

126 ## 2.3 TEMPLATE ANOMALY DETECTION MODULE

127 ### 2.3.1 SEMANTIC EMBEDDING

128 Sentence-bert (Reimers and Gurevych, 2019) is chosen as the modeling tool for semantic embed-
129 ding. Sentence-bert is a variant of BERT, and it has great advantages in computational speed com-
130 pared to the traditional Bert. The Sentence-bert uses pairwise comparison to quickly obtain the
131 embedding of sentences. In the pooling stage, the token mean or max or other criteria can be used.
132 In general, Sentence-bert greatly improves the operational speed of obtaining the sentence embed-
133 ding information, while retaining semantic information.

134 In the AbnormalLog algorithm, we first need to semantically embed the existing $n$ templates
135 into a template set $\Lambda^T = \{X_1^T, \ldots, X_n^T\}$ and generate a template semantic vector set $\Omega^T =$
136 $\{(X_i^T, \phi_i)|i = 1, \ldots, n\}$ through a pre-training procedure with template vectors and semantic vec-
137 tors matched one by one. The pre-training process of the template semantic vector set $\Omega^T$ is as
138 follows. First, we map all the template information in the template library into the semantic vector
139 set $\{\phi_1, \ldots, \phi_k, \ldots, \phi_n\}$ through the model $Q(\cdot)$, which is the Paraphrase-Multilingual-MiniLM-
140 L12-v2 model based on Sentence-bert proposed by Lab (2021) (Note: if possible, using a large
141 amount of log data to perform the pre-training is suggested). The input here is a collection of all $n$
142 log templates, and the output is a multidimensional semantic vector sets $\{\phi_1, \ldots, \phi_k, \ldots, \phi_n\}$.

$$\{\phi_1, \ldots, \phi_k, \ldots, \phi_n\} = \text{Sentence-bert}\Big(\{X_1^T, \ldots, X_k^T, \ldots, X_n^T\}\Big),$$

143 where, $\phi_i$ is the semantic vector for the $i^{th}$ log template $X_i^T$, n is the total number of templates, and
144 Q is the Sentence-bert model that maps the template set to the vector set. The dimension of semantic
145 vector $\phi_i$ is determined by Sentence-bert model. For any new log data generated at time $t$, use the
146 template $X_t^T$ to find the corresponded semantic vector $\phi_t$ in $\Omega^T$. That is

$$\phi_t = \sum_{i=1}^{n} \phi_i \times I(X_t^T = X_i).$$

147 If $(X_t^T, \phi_t) \notin \Omega^T$, put $X_t^T$ into the template set $\Lambda^T$. Then retrain the model to update $\Omega^T$.

148 ### 2.3.2 TEMPLATE ANOMALY STATUS EVALUATION

149 In the template anomaly detection, the model used to analyze log time series data needs to have the
150 ability to process sequential data. RNN related models or Transformer are all capable to process se-
151 quence data. Therefore, the Template Anomaly Detection module is a set of deep learning detection
152 framework based on LSTM and Attention mechanism. Since the sequence length of the log data
153 is limited in a few words, we adopt double-layer bidirectional LSTM stack as the core algorithm
154 structure for our template anomaly detection algorithm to capture the sequence characteristics. The

bidirectional LSTM can capture the forward sequence information, as well as the feature information from the inverse direction. The stacking model structure can improve the learning ability of the model by increasing the model structure depth. In this work, a two-layer network is built to improve the complexity of the model structure and increase the number of effective model parameters, which improve the expressive effect of the model. Also, since the bidirectional double-layer LSTM stack model has outputs in both directions, the bidirectional LSTM will splice the two outputs together. Then the spliced vectors are weighted by a layer of Attention. Then the results will be projected to the 2-dimensional space through a Full Connection layer for classification. Finally, a SoftMax layer is used to calculate the final classification probability.

Figure 1 is the framework of the log template anomaly detection module. FC is the Full Connection layer. $X_t$ is the semantic vector of the $t^{th}$ log template. $Hkl_t$ is the hidden state of the output of the $t^{th}$ LSTM module in layer $l$. $k = 1$ represents the forward LSTM, and $k = 2$ represents the reverse model. Similar to the above notation, $Ckl_t$ is the cell state output of the $t^{th}$ LSTM module in layer $l$. $k = 1$ represents the forward LSTM, and $k = 2$ represents the reverse model. $Y_t$ is the output of the $t^{th}$ sequence, which is the splicing result of two outcomes from the bidirectional LSTM module. The Attention layer processes the output from the LSTM stack. In the Attention layer,

$$u_i = Tanh(W_w Y_i), \quad \alpha_i = \frac{exp(u_i^\top u_w)}{\sum_i exp(u_i^\top u_w)}, \quad \theta = \sum_i \alpha_i Y_i,$$

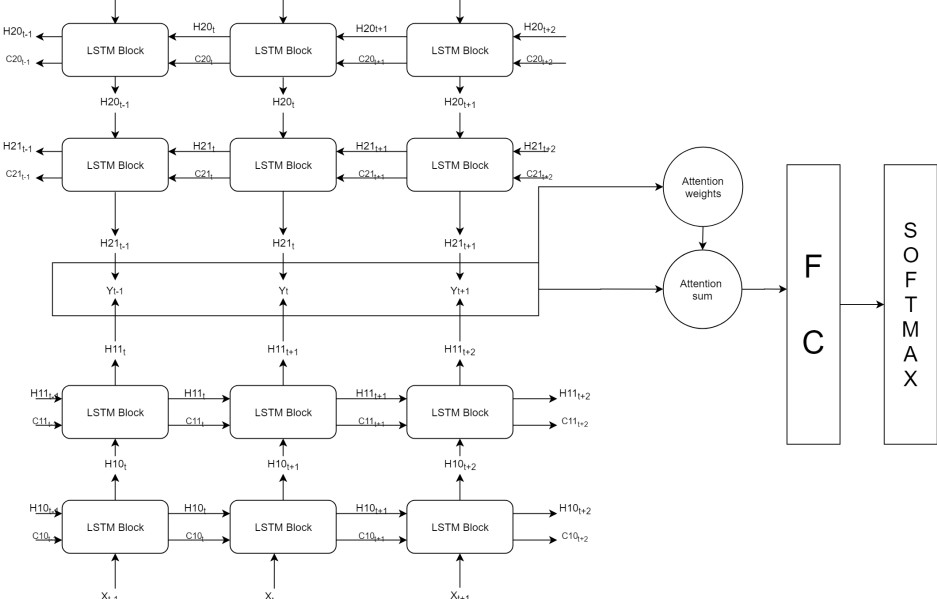

Figure 1: The framework of the Log Template Anomaly Detection Module

where $u_i$ is the output vector of the hyperbolic tangent activation function $Tanh$, with each of its element $u_{ij} \in [-1, 1]$. $\alpha_i$ is the weight of sequence $Y_i$. $\theta$ is the output of the Attention layer. $W_w$ is the weight projection matrix. $u_w$ is the sequence weight adjustment vector. Plug the output of the Attention layer into the Full Connection layer, and finalize the classification probability result through a Softmax classification layer.

For example, plug the log sequence data obtained at time $t$ into a size $s$ analysis window. At this moment, the window contains $s$ different log sequences $\{X_{t-s+1}, \ldots, X_t\}$. Next, we use the first $s - 1$ log sequences information $\{X_{t-s+1}, \ldots, X_{t-1}\}$ to predict the abnormal status of the last log sequence $X_t$. The overall operation process of the model is as follows. We first convert all log sequences in the analysis window into semantic vectors based on semantic analysis module and plug the results into the template exception detection model. Initialize the LSTM stack by initializing the hidden state H and cell state C in a random fashion. Then splice the positive and negative LSTM

outputs. Which is $Y_i = \{H11_{i+1}, H21_i\}$, where the dimensionality of $Y_i$ is twice that of the hidden state $H$. Then, all feature weights are learned through the Attention module. As mentioned above, $W_w$ and $u_w$ are initialized randomly. After the Attention layer, a Full Connection layer with two-dimensional output is designed to calculate the score of abnormal status of the template. Finally, the probability of the anomaly status of the template $X_t$ is computed through the SoftMax layer. The template exception detection model are optimized by minimizing the cross entropy loss $H(p, q) = -\sum_j p_j \ln q_j$, where $p_j$ is the true probability distribution of the event and $q_j$ is our predicted probability distribution.

### 2.4 Parameter Anomaly Detection Module

In the parameter anomaly detection, parameters can be distinguished as numeric parameters and character parameters. The difficulty of parameter anomaly detection is the design of the parameter exception detection scheme is a case driven study, and there is no way to setup a general detection scheme for different application scenarios. Even though there are numerical parameter appears, it may not either represent the quantity or quality. For example, "the type number of the car is 911" and "the type number of the car is 350", the digital parameters 911 and 350 are categorical variables and have no numerical significance. Therefore, it is not feasible to simply adopt the quantitative analysis method for all digital parameters. Similarly, if all parameters are treated as character data, the problem that lack of sensitivity to the numerical values will show up. In previous example, "the running time is $\star$ seconds". Generally, the value of $\star$ will be about $100$, but there will be significant difference when the value grows to $10,000$. Therefore, a universal parameter exception detection scheme may not be a reasonable choice. For different service scenarios, the business party should always design a personalized parameter anomaly detection scheme according to the characteristics of their service.

In this paper, we adopt the Isolation Forest for the anomaly detection of numeric parameters. The computational logic is quit straightforward. Based on the historical parameter information of the corresponded log template, by comparing with the threshold to judge the abnormal status of the new parameter in the target log data. For character parameters, our approach is identifying outliers based on their frequencies. These character parameters that have never appeared in any existing templates are treated as exceptions directly. Those with cumulative frequency lower than the predetermined threshold $\delta$ are also treated as exceptions. Note that the choices of $\delta$ varies in different application scenarios.

## 3 Experiment

In the empirical analysis, we compare the performance of our proposed AbnormalLog algorithm to three commonly used deep learning algorithms DeepLog, LogAnomaly and RobustLog. Among these methods, DeepLog and LogAnomaly are unsupervised methods, while RobustLog and our proposed AbnormalLog are supervised methods. We set the size of sequence analysis window $s = 10$, which means there will be 10 log sequences in the analysis window at any time $t$. We use the first 9 log sequences' abnormal informations to predict the anomaly status of the last log sequence.

In our experiments, we found that the unsupervised learning methods have two very obvious drawbacks. Section 3.2.1 shows that the unsupervised learning methods are highly depends on the hyper parameter $K$, which is the number of candidate templates with the Top-$K$ largest probabilities in the template anomaly detection procedure. The optimal value of $K$ varies greatly in different data application scenarios, and the optimization of the hyper parameter $K$ cost too much labor and time. In our experiment, after a large amount of model debugging works, we get the optimal value of $K$ for the HDFS data set is $K = 10$, and is $K = 20$ for the BGL data set. Section 3.2.2 shows that the highly duplication nature of the log data makes the test performance of the unsupervised learning methods unexpected inflated. To explore the true detection ability of these four methods, a comprehensive test is conducted on the deduplicated HDFS and BGL data sets. We compare the performance of these four methods based on several commonly used model evaluation criteria, $Precision$, $Recall$ and $F1$ score.

All experiments are performed on a Windows PC with *Intel I-7 9750cpu @ 2.60GHz* and *2.60GHz*. To avoid the influence of randomness, all the following experimental results are the average of five replicated experiments.

## 3.1 DATA PREPARATION

We conduct the experiments on two public data sets, which are the HDFS data set (Xu et al., 2009b), and the BGL data set (Oliner and Stearley, 2007)two classical log data sets, the HDFS data (Xu et al., 2009b) and the BGL data (Oliner and Stearley, 2007). In the log anomaly detection field, scholars often use these two data sets to testify the performance of their methods. HDFS is collected by Amazon, which has tens of millions of log records from different data block operation systems with unique IDs. BGL contains millions of system log records generated by the supercomputer BlueGene/L in Lawrence Livermore's National Laboratory. Both HDFS and BGL have their abnormality status labels marked by experts for all logs. Normal logs are al started with a symbol of "-", while the abnormal logs are not marked specifically.

In our experiments, for the HDFS data, we directly use the well-designed experimental framework provided by Deeplee-Afar (2020). This framework has nearly 0.57 millions logs, which are used as the HDFS experiment data in this paper. For the BGL data, we designed our own experimental framework. We extract the first 0.5 millions logs from the BGL data pool and use them as our experiment data. Then we perform our experiments based on the BGL sample data, including the extraction of log template sequences, the semantic embedding of different log templates, and the division of the training set and test set of the experiment. Finally, the total number of templates in our HDFS data set is 28 and that in BGL data set is 178. In order to properly apply the unsupervised learning algorithms, the data set has to be preprocessed. DeepLog only needs template sequence information, while LogAnomaly only needs the quantity information of template sequences. We split data into training sets and test sets as shown in Table 1. In the log anomaly detection, validation sets only contains normal logs. All models are trained on the original duplicated training sets, and tested on both the duplicated test sets and the deduplicated test sets.

Table 1: The data sets setup

| Data | Method | Trainning | Test (duplicated) | Test (deduplicated) |
|------|--------|-----------|-------------------|---------------------|
| HDFS | DeepLog | 12,000 | 563,060 | 17,095 |
| | LogAnomaly | 12,000 | 563,060 | 17,095 |
| | RobustLog | 12,000 | 563,060 | 17,095 |
| | AbnormalLog | 12,000 | 563,060 | 17,095 |
| BGL | DeepLog | 11,883 | 480,268 | 7,667 |
| | LogAnomaly | 11,883 | 480,268 | 7,667 |
| | RobustLog | 11,883 | 480,268 | 7,667 |
| | AbnormalLog | 11,883 | 480,268 | 7,667 |

## 3.2 EXPERIMENT RESULTS

### 3.2.1 CHOICE OF HYPER PARAMETER $K$ FOR UNSUPERVISED LEARNING METHODS

For these unsupervised algorithms (DeepLog and LogAnomaly), we evaluated the impact of the hyper parameter $K$ on the model performance. Figure 2 is the trace plot of $F1$ score at different values of the hyper parameter $K$. The major problem is that the performance of the unsupervised algorithms relies too much on the choice of $K$, and fluctuates greatly with respect to different $K$ values. For example, for the DeepLog on the BGL data set, when $K = 40$, the $F1$ score for the DeepLog on the BGL data set is 0.83; when $K = 50$, the $F1$ score drops sharply to 0 approximately. Besides, the selection of $K$ is not a easy work. It requires a lot of labor and time due to the repeated debugging.

### 3.2.2 EXPERIMENT RESULTS ON THE DUPLICATED DATA AND DEDUPLICATED DATA

In this section, we compare the perofrmance of four methods on both the duplicated and deduplicated HDFS and BGL data sets. For the HDFS data set, before deduplication, there are 16,838

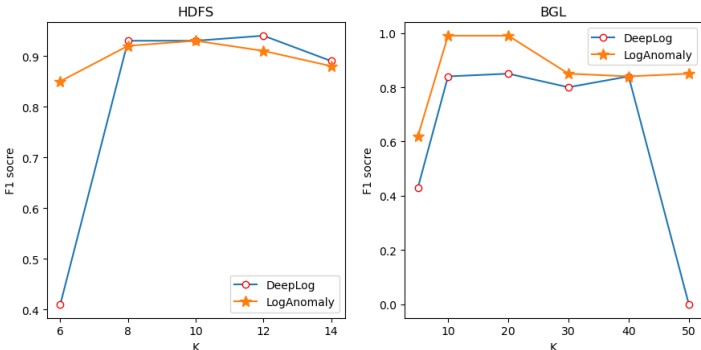

Figure 2: $F1$ score of unsupervised methods under different values of hyper parameter $K$

abnormal sequences and 553,366 normal sequences. Normal means that the sequence does not contain any exception logs. After data deduplication, there are 14,177 normal sequences and 4,123 abnormal sequences. For the BGL data set, before data deduplication, there are 285,396 normal log sequences and 206842 abnormal sequences. After data deduplication, 7,506 normal sequences and 299 abnormal sequences are obtained.

Table 2 shows the performance of unsupervised algorithm on the test data sets with severe duplication problem. The analysis results of the DeepLog, the LogAnomaly, the RobustLog and the AbnormalLog algorithms on the deduplicated HDFS and BGL data sets are summarized in Table 3. By comparing the results from Table 2 and Table 3, we can see that the test results of unsupervised algorithms are highly inflated while there is severe duplication problem. For example, on the HDFS data set, the $F1$ score decreases from 0.93 to 0.29 and $Precision$ decreases from 0.87 to 0.17 for the LogAnomaly method, which indicates that the $F1$ score of the LogAnomoly algorithm is seriously inflated by the data duplication. The similar conclusion can be also obtained for the DeepLog method.

Table 2: Performance of Unsupervised Learning Methods on Datasets with Duplications

| Data | Algorithm | Precision | Recall | F1 |
|---|---|---|---|---|
| HDFS | DeepLog | **0.92** | 0.95 | **0.94** |
| | LogAnomaly | 0.87 | **0.99** | 0.93 |
| BGL | DeepLog | 0.96 | 0.75 | 0.84 |
| | LogAnomaly | **0.98** | **1.00** | **0.99** |

From Table 3 we can conclude that supervised algorithms are significantly better than that of unsupervised algorithms with respect to the $Precision$ and $F1$ criteria. Moreover, among the four algorithms, the AbnormalLog method proposed in this work achives the highest $F1$ score with other model evaluation vriteria retain at good levels.

Table 3: Performance Comparison of Methods on the Deduplicated Data Sets

| Data | Algorithm | Precision | Recall | F1 |
|---|---|---|---|---|
| HDFS | DeepLog | 0.12 | 0.98 | 0.21 |
| | LogAnomaly | 0.17 | **1.00** | 0.29 |
| | RobustLog | **0.85** | 0.83 | 0.84 |
| | AbnormalLog | 0.82 | 0.92 | **0.87** |
| BGL | DeepLog | 0.75 | 0.90 | 0.82 |
| | LogAnomaly | 0.80 | **0.94** | 0.88 |
| | RobustLog | 0.88 | 0.82 | 0.85 |
| | AbnormalLog | **1.00** | 0.82 | **0.90** |

In summary, unlike the strong dependence of unsupervised algorithm on the hyper parameter $K$, the proposed supervised learning method AbnormalLog does not rely on any hyper parameter. There-

fore, there is no extra cost in the training process. Compared with RobustLog, which is also a supervised learning method, AbnormalLog has obvious advantages in the performance with respect to the model evaluation criteria $Recall$ and $F1$ score, except that its $Precision = 0.82$ on the HDFS data set is slightly lower than that of the RobustLog.

# 4 CONCLUSION

In this paper, we presented a new log anomaly detection algorithm, AbnormalLog. From the perspective of deep learning model architecture, AbnormalLog comprehensively uses the non-structural characteristics of log data to detect anomalies from both templates and parameters. From the empirical analysis, we demonstrate that the performance of AbnormalLog is better than three other commonly used algorithms for log anomaly detection. Particularly, AbnormalLog has the highest $F1$ score on two common data sets BGL and HDFS, and it does not rely on the hyper parameter $K$ as is the case for the unsupervised algorithms. Furthermore, based on the philosophy of our proposed algorithm, it can not only detect common exceptions in the log templates but also diagnose those customized exception patterns.

AUTHOR CONTRIBUTIONS

ACKNOWLEDGMENTS

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
