# OpenReview forum: "AbnormalLog: A Deep Anomaly Detection Method for Log Sequence Data"
_ICLR.cc/2024/Conference — ICLR 2024 Conference Withdrawn Submission_

### Official Review · Reviewer_reVm · 2023-10-30

**Soundness:** 2 fair
**Presentation:** 2 fair
**Contribution:** 1 poor
**Rating:** 3
**Confidence:** 4

**Summary:**

This paper presents a supervised learning method for log anomaly detection. However, this paper is not well-written. The technical novelties or contributions are quite limited. And only few datasets and baselines are used in the experiments.

**Strengths:**

1. The problem studied in this paper is interesting.
2. A framework has been proposed for both parameter anomaly detection and template anomaly detection.

**Weaknesses:**

1. The technical novelties or contributions are quite limited. The existing method Drain is used for log parsing, sentence-bert is chosen for semantic embedding, and double-layer bidirectional LSTM is used for template anomaly detection.
2. Many advanced log anomaly detection methods are not discussed or compared.
e.g., methods using pattern recognition and similarity comparisons (PCA, iForest, LogCluster, Invariants Mining); methods utilizing one-class objective functions (OC-SVM, OC4Seq); methods with content-awareness (Logsy, AutoEncoder).
3. Only two datasets are used in the experiments. There are some other log data available, e..g, Thunderbird.
4. The authors stated that their proposed AbnormalLog is a supervised learning method. But in practice, supervised learning based anomaly detection method is less useful. It is hard to obtain the labels. Meanwhile, the anomalies are rare compared to the large amount of normal log data. The authors did not address the challenge of the unbalanced data problem.
5. It is better to use some additional evaluation metrics such as AUC and AUPR in addition to Precision, Recall and F1.

**Questions:**

1. What are the key novelties or contributions of this work?
2. How did you address the unbalanced data issue since you model the log anomaly detection as a supervised learning problem?
3. Why proposing a supervised learning method instead of unsupervised learning?

---

### Official Review · Reviewer_edrz · 2023-11-01

**Soundness:** 1 poor
**Presentation:** 2 fair
**Contribution:** 2 fair
**Rating:** 3
**Confidence:** 4

**Summary:**

The paper presents an integrated anomaly detection system for computer system logs. This demonstrates that utilizing all the available information from logs improves the performance.

**Strengths:**

The integrated method will be of interest to practitioners

**Weaknesses:**

1. The technique presented here combines existing techniques in ways that are not novel. It also does not present any new insights. We only get to validate that performance can be improved by: (a) adding more information, (b) adding supervision.


2. Lines 228-229 "... Section 3.2.2 shows that the highly duplication nature of the log data makes ..." -- If in real world applications the log data has a high amount of duplication, then the test data should reflect the same scenario. Sanitization by deduplication is inappropriate in such cases and biases against algorithms that handle duplicated data well.


3. Line 282 "... algorithms are highly inflated while ..." -- Incorrect to say 'inflated' when the data is unbiased real world data. Instead, it should be discussed why the proposed algorithm (AbnormalLog) does not perform well when there are duplications.


4. Lines 216-217 "... DeepLog and LogAnomaly are unsupervised methods, while RobustLog and our proposed AbnormalLog are supervised methods." -- It is unfair to compare unsupervised algorithms with supervised ones at the same level. It is good to have results of unsupervised methods as they provide important information, however, more supervised methods are needed as baselines.


5. Line 150 -- Given the popularity of transformers, it would be more interesting if results were presented with transformers as well in addition to LSTM.


6. Too many typos and grammatical errors. A few examples:

  - lines 41, 42
  - line 49 -- detect => detects
  - line 57 -- extract => extracts
  - line 62 -- contains => contain
  - line 71 -- nature language processing => natural ...
  - Algorithm 1 Step 1: " ... get the their and templates ... "
  - line 142 -- sets => set
  - line 195 -- "... numerical parameter appears, it..." => "... numerical parameter, it.."
  - line 207 -- quit => quite
  - line 208 -- corresponded => corresponding, "by comparing" => "compare"
  - line 227 -- "... model debugging works," => "... model debugging,"
  - line 245 -- al => all
  - line 271 -- perofrmance => performance

**Questions:**

1. Line 94 -- Z_t data type has not been defined. Is it a binary {0, 1} variable? It could be confusing to understand whether it is a vector of {0, 1} or a real value.


2. Section 2.3.1 -- The pre-training process described here is incomplete. What is the loss / objective that is used for training?


3. Algorithm 1, Step 1 -- How is 'n' determined?

---

### Official Review · Reviewer_jFMJ · 2023-11-05

**Soundness:** 1 poor
**Presentation:** 3 good
**Contribution:** 2 fair
**Rating:** 5
**Confidence:** 4

**Summary:**

This paper proposes a method (AbnormalLog) for detecting anomalies in log sequence data using a semantic parsing technique which uses template and parameter data. The proposed method is evaluated for 2 public datasets viz. HDFS and BGL against 3 state of the art log anomaly detection techniques viz. DeepLog, LogAnomaly, RobustLog.

**Strengths:**

1. The paper addresses a relevant topic of research. Anomaly detection in logs is critical to AIOps and although this is a well-researched topic, there is still enough room for innovation in this area. Specifically, the question of how parameters can be better leveraged for anomaly detection is still an active area of research and have not been widely adopted in the industry.

2. Related work section is sufficiently detailed and most of the major log anomaly detection works have been referred starting from unsupervised clustering techniques to state of the art deep learning based methods.

3. The authors have considered both the structural characteristics as well as the parameter information in their models for anomaly detection. While the analysis of both keyword and parameter information for detecting log anomalies is nothing new the method described in sections 2.3.1 and 2.3.2 has some novelty.

**Weaknesses:**

1. Section 2.4 does not seem to have sufficiently detailed information about the detection technique. Firstly it is not clear what the authors mean by character parameters (could be words or alphanumeric values). I also didn’t quite understand why a simple frequency based approach will work for parameters. It is quite possible that some parameters are more popular than others (e.g. user "admin" vs "john doe" or port "3306" vs "3309") but this does not imply that the rare ones are anomalies. Also some parameters (like URLs etc.) will appear once or twice in the log but does not necessarily imply an anomaly.
In my opinion, for this approach to work, the first thing that needs to be done is a robust approach of segregating keywords from parameters which can be done using specialised embedding techniques. Once that is done there needs to be a method to identify normal ranges for each parameter value (and position within the template). It is not at all clear how the proposed method addresses these concerns.

2. Categorisation of parameters into numeric and character based seems too simplistic. Parameters could also be a combination of characters and numbers (alphanumeric e.g. servernames), of variable lengths (e.g. a json object as parameter) and so on. How will the frequency method work on a json object for example?
The paper has mentioned that they use Drain to identify the parameters which may not be good enough  (Drain does not work very well on variable length parameters or long temples e.g. conversational logs - as per my experience). There is prior art in the area of keyword-parameter segregation but the paper does not cite any of these prior arts. Since the authors mention parameter based detection as a novelty, some analysis of prior work in this area should have been done.

3. The experimental results have been shown for only 2 public datasets, which in my opinion, is not sufficient. The nature of the templates and parameters vary widely across logs and unless the system is tested exhaustively with various datasets and for large volumes of logs it is difficult to ascertain the efficacy of this algorithm. The dataset size was less than 1 million logs with the first dataset having only 28 templates. I think more rigorous experimentation is needed to confirm the results.

4. The experimental results do not elaborate the characteristics of the datasets viz. what was the duration of each type of log dataset, average template length, the key-parameter ratio in the templates etc. It is also not clear what constitutes a normal and an abnormal sequence in the logs. Some explanation around different types of anomalies detected and the breakup of accuracy on the different anomaly types would have been helpful for this review.

5. One of the most important considerations in log anomaly detection is the question of scaling, i.e. how well the system scales with high volume of logs, what voume of logs can be processed per day and how much real time we can get in the detection of anomaly. There is no mention of the paper regarding this aspect and as such we have no idea whether this system is deployable in a real life scenario with a high volume of incoming logs. This is probably the biggest drawback of the paper.

**Questions:**

Please refer to the weaknesses section for the questions.